# Prognostic Significance of Glycolysis-Related Genes in Lung Squamous Cell Carcinoma

**DOI:** 10.3390/ijms25021143

**Published:** 2024-01-17

**Authors:** Sultan F. Kadasah

**Affiliations:** Department of Biology, Faculty of Science, University of Bisha, P.O. Box 551, Bisha 61922, Saudi Arabia; sukadasah@ub.edu.sa

**Keywords:** lung squamous cell carcinoma, glycolysis, signature model

## Abstract

Lung squamous cell carcinoma (LUSC) is one of the most common malignancies. There is growing evidence that glycolysis-related genes play a critical role in tumor development, maintenance, and therapeutic response by altering tumor metabolism and thereby influencing the tumor immune microenvironment. However, the overall impact of glycolysis-related genes on the prognostic significance, tumor microenvironment characteristics, and treatment outcome of patients with LUSC has not been fully elucidated. We used The Cancer Genome Atlas (TCGA) dataset to screen glycolysis-related genes with prognostic effects in LUSC and constructed signature and nomogram models using Lasso and Cox regression, respectively. In addition, we analyzed the immune infiltration and tumor mutation load of the genes in the models. We finally obtained a total of glycolysis-associated DEGs. The signature model and nomogram model had good prognostic power for LUSC. Gene expression in the models was highly correlated with multiple immune cells in LUSC. Through this analysis, we have identified and validated for the first time that glycolysis-related genes are highly associated with the development of LUSC. In addition, we constructed the signature model and nomogram model for clinical decision-making.

## 1. Introduction

Lung cancer seems to be the major cause of cancer deaths globally [1]. Non-small cell lung cancer is the most common histological form of lung cancer, accounting for more than 85% of all occurrences, and is characterized by high metastasis and recurrence, including the following four subtypes: lung adenocarcinoma, lung squamous carcinoma (LUSC), large cell carcinoma, and pulmonary neuroendocrine carcinoma, with LUSC accounting for approximately 30% of all lung cancer cases [2].

LUSC is treated in a similar way to the vast majority of solid tumors. The main treatment modalities currently include surgery, chemotherapy, radiotherapy, immunotherapy, and targeted therapy. Despite significant advances in lung cancer prevention, diagnosis, and therapy, the 5-year overall survival rate for lung cancer patients is just 19% [3,4]. Glycolysis is quite important for the growth of certain malignant tumors, as it can provide the energy required by active cancer cells to promote the growth of such cancer cells and other important tumor behaviors [5]. During this process, glucose will be broken down into ethanol and aldehydes, which will provide the energy and molecular structure needed for other intracellular substances that contribute to the growth and development of tumors [6]. Therefore, in this study, we determined the risk genes and their correlation coefficients associated with the progression of LUSC based on the TCGA LUSC database [7] and established a prognostic model for LUSC based on various clinical indicators, thus providing a basis for early intervention in LUSC patients.

## 2. Results

### 2.1. Identification of DEGs with Prognostic Effects

A total of 5676 DEGs were obtained from our study. 2288 genes upregulate, while 3388 genes upregulate (Figure 1A).

The DEGs were enriched in the p53 signaling pathway, skin development, the cGMP-PKG signaling pathway, second-messenger-mediated signaling, and other processes (Figure 1C).

### 2.2. Signature Model Construction

A total of 109 glycolysis-associated DEGs were eventually obtained. After dimensionality reduction based on lasso, a model including four genes (*AGL*, *ALDOA*, *ADH1B*, and *ALDH3B1*) was obtained (Figure 2A,B). Riskscore = (−0.0045) × *AGL* + (0.0633) × *ALDOA* + (0.0048) × *ADH1B* + (0.0937) × *ALDH3B1*, where lambda.min = 0.0523 and the KM curve shows that log-rank *p* < 0.01 indicates a significant difference between the survival times in the high Riskscore score group and the low Riskscore score group (Figure 2D). The ROC curves of the model with AUC = 0.53 at 1 year, AUC = 0.62 at 3 years, and AUC = 0.66 at 5 years indicated that the risk model was a good predictor of squamous lung cancer (Figure 2). In addition, we found positive correlations with macrophages, endothelial cells, and NK cells and negative correlations with uncharacterized cells (Figure 3 and Figure 4).

### 2.3. Construction of a Nomogram Model

We developed a line plot capable of predicting OS at 1, 3, and 5 years using the expression of glycolysis-related DEGs and other clinical features (including age, gender, and TNM stage) in squamous lung cancer. *AGL*, *ALDOA*, and *ADH1B* were eventually included in the nomogram model (Figure 5). The column line plot has a C-index of 0.63. To read the column line plot, a vertical line should be drawn up to the top dotted row, assigning points to each variable. The total points for the patients can then be summed, and the probabilities of 1-, 3-, and 5-year OS can be obtained by plotting vertical lines from the total points rows. Calibration plots of the 1-, 3-, and 5-year OS probabilities show good agreement between the OS predicted by the column line plots and the actual OS of patients with squamous lung cancer.

### 2.4. Immunological Profile and Tumor Mutational Load Analysis

The findings of the immunological infiltration demonstrated a positive correlation between Purity and the expression of *AGL* and *ALDOA*, and a negative correlation between Purity and *ALDH3B1*. Whereas the expression of *AGL* was positively connected with CD4+ T cells, macrophages, and neutrophils, the expression of *ALDOA* was negatively correlated with B cells, CD8+ T cells, CD4+ T cells, macrophages, neutrophils, and dendritic cells. Purity and ADH1B expression were negatively connected, whereas B cell, CD8+ T cell, CD4+ T cell, macrophage, neutrophil, and dendritic cell expression were positively correlated (Figure 6).

Tumor mutation load data revealed a negative correlation between *ADH1B* expression and TMB but no correlation between *AGL*, *ALDOA*, or *ALDH3B1* expression and TMB (Figure 7).

### 2.5. Validation of the Glycolysis-Related Genes in Tissue Samples

To confirm the gene signature’s reliability, we used immunohistochemical data to detect protein levels in four genes in normal lung tissue samples and lung cancer cell lines. The results showed that *ALDOA*, *ADH1B,* and *ALDH3B1* were significantly overexpressed in tumor samples compared to normal samples (Figure 8).

## 3. Materials and Methods

### 3.1. Data Acquisition

The Cancer Genome Atlas (TCGA), a comprehensive and publicly accessible database, was utilized as the primary source for gathering data on tumor samples. Specifically, this research incorporated information from 501 distinct tumor samples, each providing valuable insights into various cancerous conditions. In addition to the TCGA, this study also employed data from the Genotype-Tissue Expression (GTEx) database. GTEx is known for its rich collection of normal tissue profiles, providing a crucial comparison point for understanding abnormal or disease states. For this research, 578 normal lung tissue samples from the GTEx database were meticulously analyzed. These samples are vital for establishing a baseline of normal genetic expression, which, when compared to the cancerous samples from TCGA, aids in identifying specific genetic alterations and expressions associated with lung cancer. This comparison is essential for understanding the genetic basis of cancer and for identifying potential therapeutic targets.

### 3.2. DEG Identification

To explore differential mRNA expression, the study utilized the Limma package within the R software (4.0.5 version). Limma, known for its robust statistical methods, facilitated the identification of significant changes in mRNA levels. The criteria set for discerning noteworthy differential expression were stringent: only those with a false discovery rate (FDR) less than 0.05, combined with a log2 fold change greater than 1 or less than −1, were considered significant. This dual-threshold approach ensured that the findings were both statistically valid and biologically meaningful.

### 3.3. Enrichment Analysis

The obtained DEGs with prognostic effects were analyzed for GO and KEGG pathways using the clusterProfiler package and the ggplot2 package of R software (4.0.5 version) to explore the cellular localization of these genes in squamous lung cancer and their involvement in biological processes and signaling regulatory networks.

### 3.4. Signature Model Construction

Based on RNA-seq information from TCGA and clinical samples, the best model was chosen by the least absolute shrinkage and selection operator (LASSO) regression algorithm using 10-fold cross-validation. The correlation between the signature prognostic model’s RiskScore and various immune cells was analyzed. We used the RiskScore scores obtained from the model to explore the correlation with immune cells.

### 3.5. Nomogram Model Construction

Initially, single- and multivariate cox modeling analyses were run, and forestplots were performed to represent each variable (*p*-value, HR, and 95% CI) using R’s forestplot software package (4.0.5 version). Column line plots were created using the RMS software program (Version 22) according to the results of the multivariate Cox proportional risk analysis to estimate the overall recurrence rates of patients with squamous lung cancer at 1 year, 3 years, and 5 years. The column line graphs depict these characteristics graphically and allow the prognosis risk of individual patients to be determined using the points associated with each risk factor.

### 3.6. Immune Infiltration and Tumor Mutational Load Analysis

To examine the connection between signature’s genes and the immunological microenvironment, we obtained the immune infiltration of these genes from the TIMER database, and, in addition, we explored their relationship with the tumor mutational load, which we characterized using Spearman’s correlation analysis and visualized using the ggstatsplot package of R software (4.0.5 version).

### 3.7. Immunohistochemistry

Immunohistochemistry staining results were extracted from the Human Protein Atlas “https://www.proteinatlas.org/ (accessed on 17 September 2023)”. The expression of glycolysis-related genes was compared in lung squamous cell carcinoma samples and normal lung tissue.

## 4. Discussion

Squamous lung cancer is one of the deadliest malignancies in the world [8]. The overall prognosis of patients with squamous lung cancer continues to be poor despite recent improvements in diagnosis and therapy, with one of the primary causes being the absence of useful prognostic indicators. As a result, it is critical to investigate relevant prognostic indicators and treatment targets for LUSC. Because of the variability and complexity of the tumor immune microenvironment, immunotherapy benefits only a tiny percentage of patients [9,10,11,12]. Glycolysis underlies the proliferation and differentiation of malignant tumors, provides energy for tumor metabolism, and is considered to be an important factor contributing to tumors [13]. Currently, many studies have been published on the mechanisms of glycolysis in LUSC, but no application of glycolysis-related genes has been investigated [14]. In the first step of the study, we used the data in TCGA to screen for differential genes in squamous lung cancer. We performed GO and KEGG analyses on these genes, and the results showed that their enrichment results were mainly in the p53 signaling pathway, skin development, the cGMP-PKG signaling pathway, second-messenger-mediated signaling, and other processes. After collecting 293 glycolysis-related genes from the MSigDB database and intersecting them with the previously obtained differential genes, we enriched the 109 genes obtained again for analysis. These genes were then used to construct a signature and nomogram model, respectively. Our signature model included a total of four genes, and the ROC curves of the model had AUC = 0.53 at 1 year, AUC = 0.62 at 3 years, and AUC = 0.66 at 5 years, indicating that the risk model is a good predictor of squamous lung cancer and has some application. In addition, our study found a positive correlation between the model and the presence of multiple immune cells. In our study, we also constructed a nomogram model including *AGL*, *ALDOA*, *ADH1B*, age, gender, and TNM-stage, which has good performance in predicting the prognosis of patients with LUSC. To investigate the mechanism of the genes in the model, we additionally analyzed the effect of *AGL*, *ALDOA*, *ADH1B*, and *ALDH3B1* expression on the level of immune infiltration in LUSC. TMB refers to the number of non-synonymous mutations in a given genomic region of the somatic cells, usually expressed as the number of mutations per Mb and, in earlier studies, also directly as the number of mutations. The expression of *ADH1B* is negatively correlated with TMB and may be a new target for tumor immunotherapy. This may give us new ideas to explore how glycolysis-related genes regulate pathway proteins to influence the level of tumor immune infiltration.

*AGL*, primarily known for its role in glycogen breakdown, has been identified as a significant factor in bladder cancer. It is recognized as a biomarker that inhibits tumor growth in this type of cancer [15]. However, the reduction or silencing of AGL activity has been observed to facilitate the growth of bladder tumor cells through various pathways. These include the enhancement of glycine synthesis and the activation of HAS2-driven hyaluronic acid (HA) production [16]. Moreover, recent research has extended the understanding of AGL’s role to NSCLC. In this context, it has been proposed that decreasing AGL activity similarly accelerates the growth of NSCLC cells, a process also influenced by HAS2 [17].

*ALDOA*, when overexpressed, has been associated with increased proliferation and metastasis in lung cancer cells [18]. Furthermore, research by Zhang et al. indicates a correlation between heightened *ALDOA* transcription levels and genes related to the cell cycle, suggesting *ALDOA*’s potential role in regulating the progression of non-small cell lung cancer [19]. Nonetheless, the link between *ALDOA* expression levels and both the prognostic outcomes and the extent of immune infiltration in lung adenocarcinoma remains unexplored.

Several genes belonging to the ADH family exhibited statistically significant differences in expression between tumors and neighboring normal tissues. The levels of *ADH1B* expression exhibited a notable and consistent reduction in tumor tissue. Reduced ADH1B expression correlated with a poorer overall survival outcome, and the combined impact of these genes yielded a more significant prognostic value than the cumulative effects of each gene separately in the TCGA database. The levels of *ADH1B* transcripts were associated with the control of metabolism, cell cycle, DNA repair, and pathways related to cancer. A prognostic risk score model was created to forecast the prognosis of LUAD, demonstrating effective performance in anticipating overall survival (OS) at 1, 3, and 5 years. In line with findings from TCGA, our dataset with 111 patients in replication demonstrated that the depression in tumor expression was most notable for *ADH1B*. Furthermore, lower expression of *ADH1B* was correlated with the occurrence of vascular, pleural, and lymphatic invasions. Moreover, smoking status and/or cumulative smoking history do not provide a direct, clinically accessible proxy for tumor *ADH1B* expression.

Studies have validated that the expression of *ALDH3B1* is elevated in lung adenocarcinoma tissues compared to normal tissues. This expression has an important impact on the prognosis of other cancers, such as lung adenocarcinoma. A widely accepted idea is that the activity of *ALDH3B1* influences the processing of aldehydes, such as acetaldehyde, within the metabolic system [20]. Aldehydes have stimulatory effects in humans and induce mutations that lead to cancer [21].

However, there are still many unknowns that limit the clinical application of immunotherapy, and despite our comprehensive and systematic analysis of these glycolysis-associated genes and the construction of the Signature model and nomogram model, there are still limitations to this study. From a modeling perspective, an AUC of 0.66 is not ideal. There could be other confounding factors or other genes that could be related. We need more comprehensive multicenter data on patients with squamous lung cancer to validate our model with accuracy and improve the credibility of our results.

## Figures and Tables

**Figure 1 ijms-25-01143-f001:**
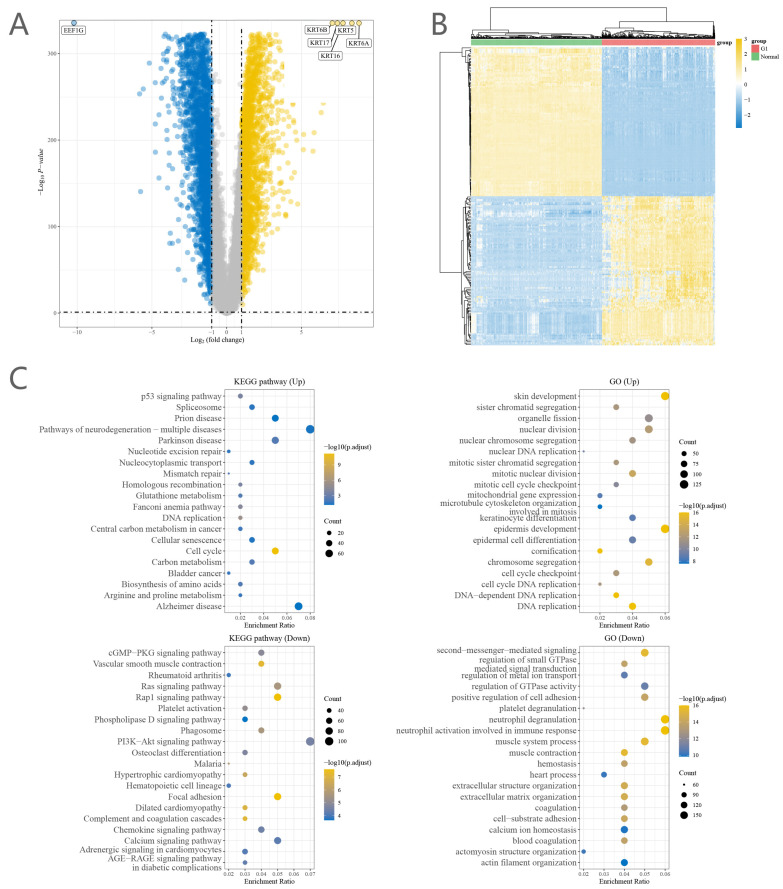
(**A**) Volcano map of differential genes. Yellow represents upregulation, blue represents downregulation. (**B**) Heat map of clustering of differential genes. (**C**) Enrichment analysis of differential genes.

**Figure 2 ijms-25-01143-f002:**
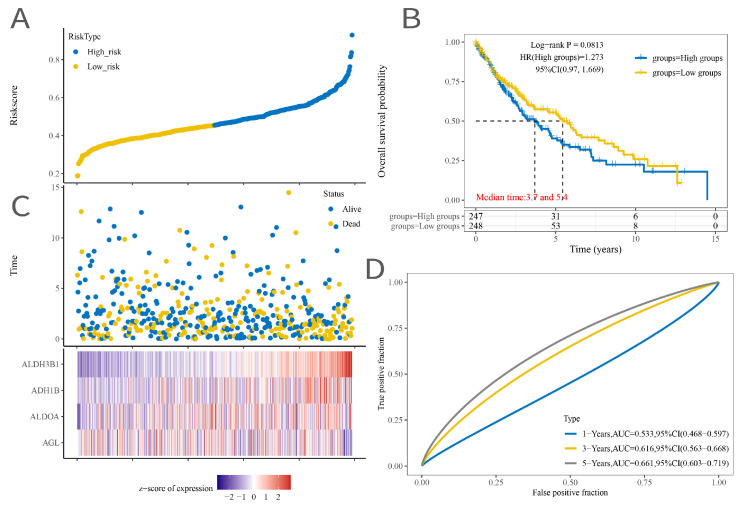
(**A**) Distribution of high- and low-scoring samples. (**B**) KM curves for the high expression group combined with the low expression group. (**C**) Distribution of sample scores. (**D**) ROC graph.

**Figure 3 ijms-25-01143-f003:**
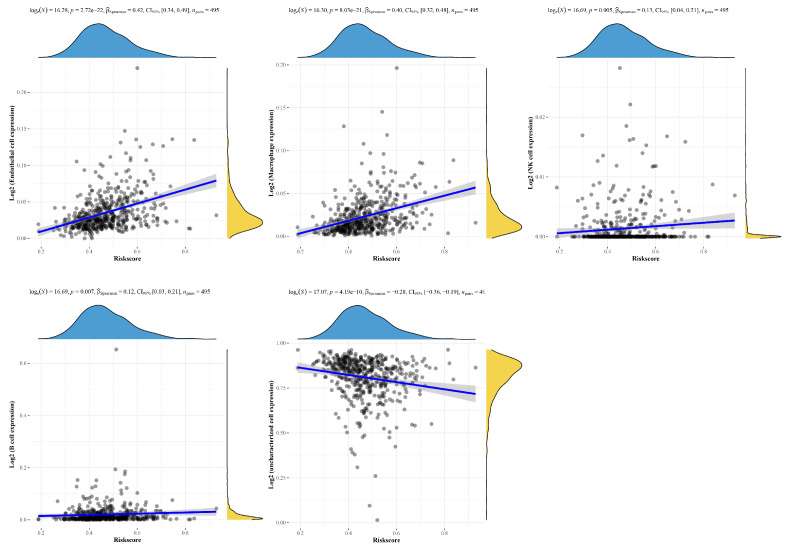
Correlation analysis of signature models and immune cells.

**Figure 4 ijms-25-01143-f004:**
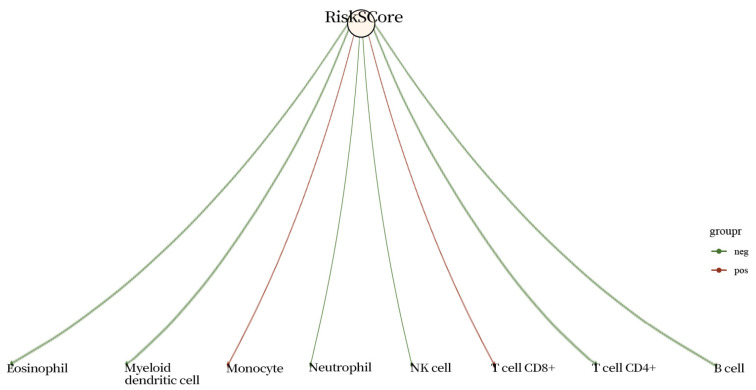
Correlation analysis of signature models and immune cells using CIBERSORT.

**Figure 5 ijms-25-01143-f005:**
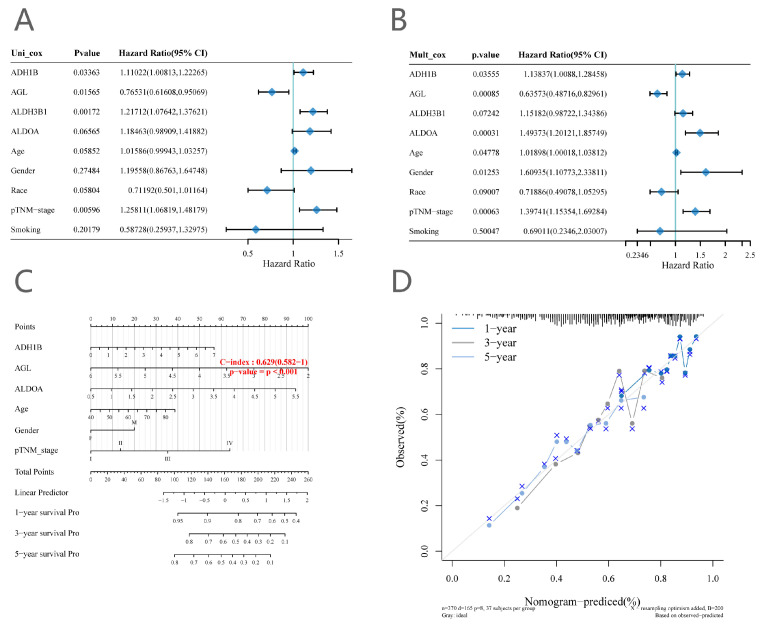
(**A**) Univariate Cox regression. (**B**) Multivariate Cox regression. (**C**) Nomogram model. (**D**) Determine how well the model fits the real situation; the closer to a straight line, the better the model predicts.

**Figure 6 ijms-25-01143-f006:**
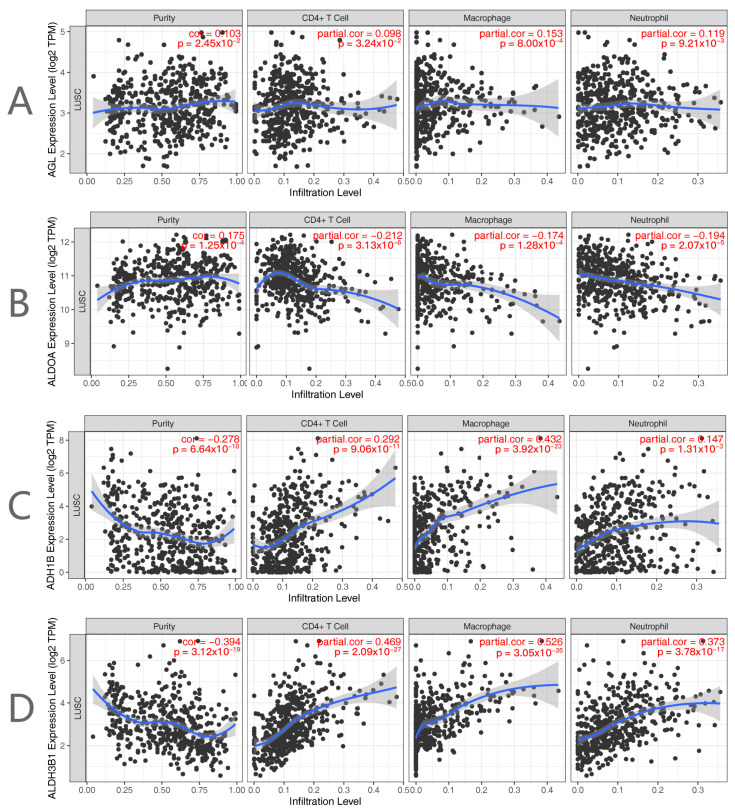
Levels of immune infiltration of genes in the signature. (**A**) *AGL*, (**B**) *ALDOA*, (**C**) *ADH1B*, (**D**) *ALDH3B1*.

**Figure 7 ijms-25-01143-f007:**
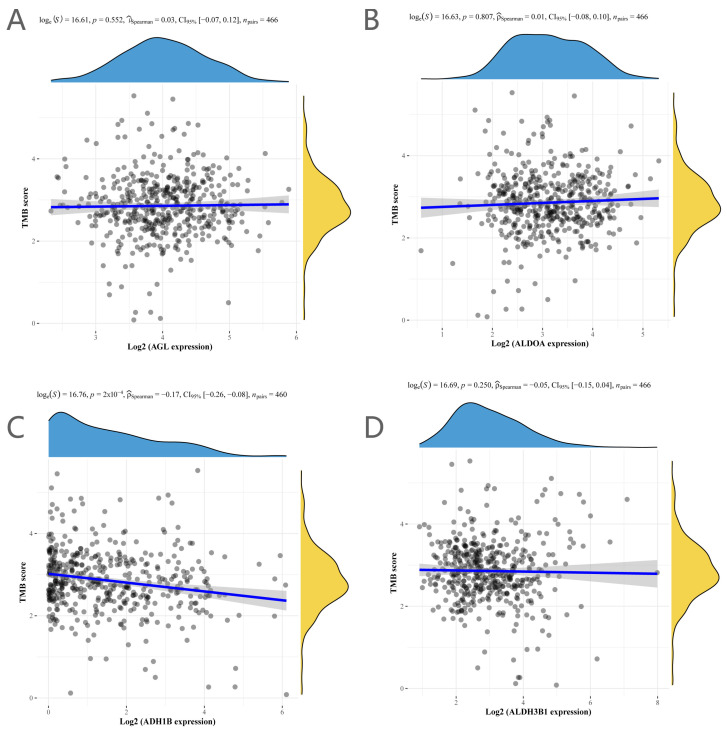
TMB levels of genes in the signature. (**A**) *AGL*, (**B**) *ALDOA*, (**C**) *ADH1B*, (**D**) *ALDH3B1*.

**Figure 8 ijms-25-01143-f008:**
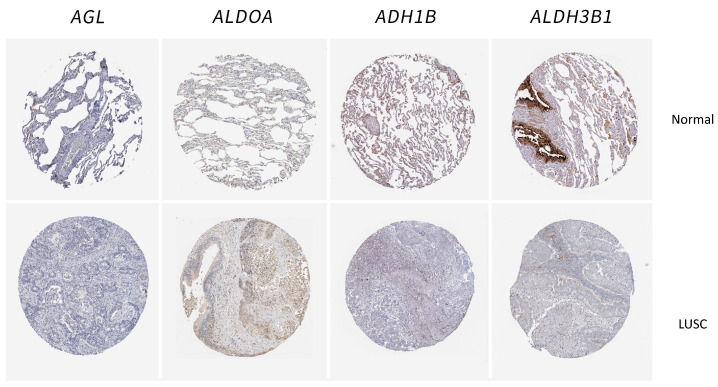
Validation of the gene by immunohistochemistry between LUSC and normal samples.

## Data Availability

The datasets supporting the conclusions of this article are available in the Genotype-Tissue Expression (GTEx) database and the cancer genome atlas (TCGA) database.

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
