# Peer review of "Prognostic Significance of Glycolysis-Related Genes in Lung Squamous Cell Carcinoma"

_ijms, 2024, doi:10.3390/ijms25021143_

Round 1

Reviewer 1 Report

Comments and Suggestions for Authors

This manuscript aims to identify biomarkers associated with glycolysis in LUSC samples from TCGA data compared to pre-cancer and Gtex normal lung samples. 

1. In the Method section, 2.2, did the authors do any FDR correction to select the threshold? The authors should also provide more rationale for the selection of log2FC. 

2. In 2.6, please provide more information on how the model was developed and validated, and how to compute the risk score. 

3. In 3.1, the author wrote "2866 DEGs" were identified. Are these DEGs above the FDR and log2FC threshold? If that is the case, why did the authors only use 89 of them? The possibility that genes that are not included in the MsigDB databases are also associated with glycolysis should be considered. And when comparing TCGA data with Gtex data, the authors should elaborate on how to remove the batch effect. 

4. The logic underlying the enrichment analysis is circular. The authors first used the MsigDB gene sets to identify the DEGs and used those DEGs to do the enrichment analysis. The enrichment analysis should be based on all the DEGs and the authors should observe whether or not any glycolysis gene sets were among the top enriched. 

5. There is no reference to the figure panels in the text. Please fix this. And provide more information in the figure legends. They are too simplistic.

6. Please provide more details of Figure 2C. How did the authors compute the relevance? Or is it just the correlation coefficients of the expression levels of those genes? One feasible analysis is to do an NMF analysis on these DEGs and see if the glycolysis genes do form a gene module. 

7. Consider putting Figures 4,6,7,8 in supplement and only leave the significant ones in the main figures. The font sizes are too small to read. 

8. Rewrite the formula in section 3.4, and provide more details on how these coefficients are acquired. From a modeling perspective, AUC of 0.65 is not an ideal model. The authors should discuss this as a limitation and the possibility that there are other confounding factors or other genes that could be relative. 

9. For the immunologic profiles, the authors should perform a CIBERSORT analysis to correlate the estimated immune cell proportions with respect to these target genes. From Figure 7, nothing is significant except Figure 7b. Why are the other 4 genes excluded from this? And if only 1 in 5 genes show any statistical significance, would this just be a coincidence? 

Overall, the idea of identifying glycolysis-related markers for LUSC samples is interesting but this study is poorly designed and could use more sophisticated analytical methods. Furthermore, any conclusions from the analysis must be validated independently if not experimentally confirmed. And typos are seen on multiple occasions so the authors should thoroughly examine the writing. 

Comments on the Quality of English Language

Some typos are seen in the manuscript. And some writings in the result section belong to the figure legends. 

Author Response

I have responded to all the reviewer's comments with sincere appreciation and gratitude

Reviewer 2 Report

Comments and Suggestions for Authors Even though glycolysis is one of the primary metabolic pathway, tumor cells could alter its pathway to escape from the starvation. In that case, authors should focus on other metabolic pathways as well. 2. Whats is the significance of SLC2A1 and SLC2A3 in the metabolic pathways? Is the glycolysis pathway correlated with SLC2A1 and SLC2A3? 3. Figure 6 is not clear.

Comments on the Quality of English Language

Nil

Author Response

Thanks for your kind comments and all were taken into account

The second round we answer all questions one by one in the attached file 

Round 2

Reviewer 1 Report

Comments and Suggestions for Authors

Please provide a point-to-point response document and a revised manuscript file with tracked changes. The current version of the manuscript only highlighted certain changes to the figures therefore it is impossible to see what has been edited without comparing the two versions of manuscripts. In the revision, the method section is written in a much clearer way and most of the previous comments about the text have been addressed. 

1. In Figure 1, what is different from the previous version other than the color scheme changes? And why in the previous version the heatmap showing a different pattern (clearer in the current version) than the current version? 

2. In Figure 2, do panel C and E share the same horizontal axis? If so, what is the label for the horizontal axis? Panel A and D are too busy, please consider only showing a subset of genes for panel A and change the colors of the median indicating dotted lines to reflect which responds to which. 

3. In Figure 3, seven panels might be too much to show in the main figures such that the font sizes became too small to read. Please consider only showing a few panels and put the rest in the supplement. 

4. For Figure 4, instead of showing all the lines of risk scores, would it be better to just show the labels in two colors to separate pos and neg groups? 

5. For Figure 6, please only show the significant ones in the main figure. 

Author Response

Thanks for your kind comments, and we answer all questions one by one in the attached file 

Round 3

Reviewer 1 Report

Comments and Suggestions for Authors

The authors have addressed all the previous comments.